# Entrainment dominates the interaction of microalgae with micron-sized objects

Raphaël Jeanneret[1], Dmitri O. Pushkin[2], Vasily Kantsler[1] & Marco Polin[1]

The incessant activity of swimming microorganisms has a direct physical effect on surrounding microscopic objects, leading to enhanced diffusion far beyond the level of Brownian motion with possible influences on the spatial distribution of non-motile planktonic species and particulate drifters. Here we study in detail the effect of eukaryotic flagellates, represented by the green microalga *Chlamydomonas reinhardtii*, on microparticles. Macro- and microscopic experiments reveal that microorganism-colloid interactions are dominated by rare close encounters leading to large displacements through direct entrainment. Simulations and theoretical modelling show that the ensuing particle dynamics can be understood in terms of a simple jump-diffusion process, combining standard diffusion with Poisson-distributed jumps. This heterogeneous dynamics is likely to depend on generic features of the near-field of swimming microorganisms with front-mounted flagella.

[1] Physics Department, University of Warwick, Gibbet Hill Road, Coventry CV4 7AL, UK. [2] Department of Mathematics, University of York, Heslington, York YO10 5DD, UK. Correspondence and requests for materials should be addressed to M.P. (email: m.polin@warwick.ac.uk).

Swimming microorganisms navigate commonly through fluids characterized by a variety of suspended microparticles, with which they inevitably interact. From malarial parasites meandering around densely packed red blood cells and infecting them[1], to protists which regulate primary production in oceans and lakes by grazing on sparsely distributed microalgae and bacteria[2,3], these interactions can have important biological and ecological implications. Some, like biogenic mixing, are still intensely debated[4–6]. Understanding the basic mechanisms of such interactions is timely, as a vast number of plastic micro- and nanoparticles of anthropogenic origin scattered throughout the world's oceans (up to $\sim 10^5\,\mathrm{m}^{-3}$) appear to be easily ingested by microorganisms, which then recycle plastics within the marine food web with currently unknown consequences[7]. From a physics perspective, these systems are particularly appealing. Abstracted as binary suspensions[8,9], where an active, self-propelled species interacts with a passive, thermalized one, they represent a naturally occurring category of out-of-equilibrium stochastic systems driven by energy produced directly within the bulk, rather than transmitted through the system's boundaries. Easily accessible experimentally, and amenable to detailed quantitative modelling[10], they are uniquely placed to provide benchmark tests for theories of out-of-equilibrium statistical mechanics[11–13]. Currently best characterized are the so-called bacterial baths. Wu and Libchaber[14], and more recently[15–18], showed that colloidal particles within bacterial suspensions perform a persistent random walk leading in bulk to a diffusivity up to $\sim 10 \times$ the thermal value[18]. This increase is proportional, at least in the dilute limit, to the product of bacterial speed and concentration, a consequence of random successive single-particle interactions[5,19–23]. The intrinsic non-equilibrium nature of bacterial baths is also known to lead to other peculiar effects, including motility-induced colloidal pair interactions[10] and coupling between enhanced translational and rotational diffusion[24].

Microparticles' interactions with the other major class of microorganisms, eukaryotic flagellates, is distinctly less explored. Almost exclusively larger than bacteria ($\sim 10$–$100\,\mu\mathrm{m}$), these species probe a new and biologically relevant physical regime, where the microparticles' size is significantly smaller than that of the microorganisms (but see also ref. 25 for the effect of bacterial motion on molecular diffusion). Working with the microalga *Chlamydomonas reinhardtii* (CR), often studied as model eukaryotic microswimmer, the pioneering study of Leptos *et al.*[26]—followed by ref. 27—reported an increase in particle diffusivity of magnitude similar to the bacterial case, as a consequence of loop-like particle trajectories induced by the far-field flow of the algae[21,22,28,29]. Particle displacements followed a diffusively scaling fat-tailed distribution which, if valid for arbitrary large observation times, would imply a violation of the Central Limit Theorem[30]. These fat-tailed distributions have indeed been observed in short simulations[21,31], but should converge to normal distributions at long times[29,31]. This prediction, however, has not been tested experimentally.

Here we revisit the behaviour of colloids within a suspension of CR, taken as representative of eukaryotic flagellates, and reveal that their overall dynamics is in fact dominated by rare but dramatic entrainment events. These jumps underpin the surprisingly large diffusivity we observe directly in both sedimentation and collective spreading experiments, $\gtrsim 40 \times$ greater than values reported previously for the same geometry. The colloids' behaviour, alternating entrainments and periods of standard enhanced diffusion, is fundamentally different from the persistent random walk common with bacterial suspensions. Through microscopic experiments, simulations and analytical modelling we show instead that this dynamics is well captured by a simple jump-diffusion model.

## Results

**Experiments**. A full description of the experimental procedures can be found in the Methods and Supplementary Methods sections. Briefly, wild-type CR strain CC125 was grown axenically in Tris-acetate-phosphate medium[32] at 21 °C under continuous fluorescent illumination. Cells were harvested in the exponential phase ($\sim 5 \times 10^6\,\mathrm{cell\,ml}^{-1}$), concentrated by gentle spinning and then resuspended to reach the desired concentration. Polystyrene microparticles (1 μm diameter) were then added to the suspension, and their diffusivity at different CR concentrations ($N_\mathrm{c}$) was measured from three different sets of microfluidic experiments (Supplementary Methods 1): mapping the particles' sedimentation profile at steady state; measuring the relaxation dynamics of an inhomogeneous distribution of particles; and by long timescale direct tracking of individual colloids' dynamics in a thin Hele–Shaw cell. Schematic representations of the experimental setups are shown Supplementary Figs 1 and 2. Except for the sedimentation experiments, the medium was density-matched with the colloids using Percoll[33]. This increased its viscosity ($\eta_\mathrm{percoll} = (1.5 \pm 0.1)\eta_\mathrm{water}$) as reflected in the slower average swimming speed, $\langle v \rangle$, of the algae. We measured $\langle v \rangle_\mathrm{S} = 81.8 \pm 3.5\,\mu\mathrm{m\,s}^{-1}$, $\langle v \rangle_\mathrm{CS} = 40.9 \pm 3.5\,\mu\mathrm{m\,s}^{-1}$ and $\langle v \rangle_\mathrm{HS} = 49.1 \pm 2.5\,\mu\mathrm{m\,s}^{-1}$ for the sedimentation, spreading and tracking experiments respectively. A colloidal Péclet number for tracer/swimmer interactions can be defined as $Pe = vL/D_0$, where $v$ and $L(\sim 10\,\mu\mathrm{m})$ are CR's characteristic speed and body length, and $D_0$ the Brownian diffusivity of the colloids. We obtain $Pe_\mathrm{S} \simeq 2{,}000$, $Pe_\mathrm{CS} \simeq 1{,}500$ and $Pe_\mathrm{HS} \simeq 1{,}750$, respectively for the three experiments.

**Macroscopic diffusion**. Macroscopic diffusion experiments coarse-grain over the colloids' microscopic dynamics and ensure the direct measurement of their effective transport properties, in the spirit of Jean Perrin's seminal work on sedimentation equilibrium[34]. We begin by characterizing the colloids' steady-state sedimentation profile at increasing CR concentrations, always within the dilute regime (volume fractions $\lesssim 0.15\%$). The average concentrations probed will always be well below the threshold required to induce bioconvective instabilities within our $\sim 200\,\mu\mathrm{m}$-thick sample cells. At the same time the concentration profile is not exactly homogeneous, but slightly decreasing with height, see Supplementary Fig. 3 and Supplementary Note 1. However, as it will be evident from the results, this inhomogeneity did not appear to have appreciable consequences on our measurements. Figure 1a shows that even when the algae are present, the microparticles' distributions are still in excellent agreement with simple exponential profiles, but crucially with different effective gravitational lengths $l_\mathrm{g,eff}$ (Fig. 1b). The exponential profiles are a standard consequence of the balance between the particles' sedimentation speed $v_\mathrm{sed}$, here due to a $\delta\rho = 50\,\mathrm{g\,l}^{-1}$ density mismatch, and their effective diffusivity, combining passive and active processes. Boltzmann-like distributions are indeed what should be expected even in these out-of-equilibrium systems, at least for small enough $v_\mathrm{sed}$ (ref. 35), akin to what has been observed for active colloids alone[36]. The characteristic length $l_\mathrm{g,eff}$, experimentally observed to be proportional to the concentration of algae, allows us to measure the concentration-dependent effective diffusivity as $D_\mathrm{eff} = v_\mathrm{sed}\, l_\mathrm{g,eff}$ (Fig. 1c, orange squares, Supplementary Fig. 4 for a close-up at low concentrations). We obtain $D_\mathrm{eff} = D_0 + \alpha_\mathrm{S} N_\mathrm{c}$, where $D_0 = 0.40 \pm 0.01\,\mu\mathrm{m}^2\,\mathrm{s}^{-1}$ is the thermal diffusivity, the algal concentration $N_\mathrm{c}$ is in units of $10^6\,\mathrm{cells\,ml}^{-1}$, and $\alpha_\mathrm{S} = 1.71 \pm 0.14\,(\mu\mathrm{m}^2\,\mathrm{s}^{-1})/(10^6\,\mathrm{cells\,ml}^{-1})$ (slopes $\alpha$ will be expressed in these units throughout the paper). Within the same experiments, however, the diffusivity can also be inferred

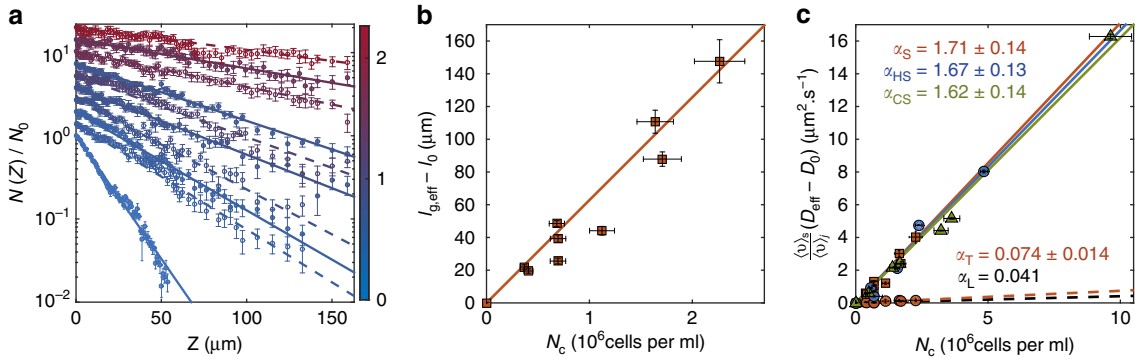

**Figure 1 | Effect of swimming algae on particle diffusivity.** (**a**) Semi-log plot of the normalized density profiles of 1 μm-PS colloids along the gravity direction for different cell concentrations. Full lines are best exponential fits to the data. Curves have been shifted apart along the $y$ axis for clarity. Colour bar: CR concentration $N_c$ in units of $10^6$ cells ml$^{-1}$. Error bars represent the s.d. (**b**) Gravitational length $l_{g,eff} - l_0$ as a function of $N_c$ extracted from the fits in **a**. The solid orange line is the best linear fit to the data, $l_{g,eff} - l_0 = ((62.8 \pm 5)N_c)$ μm ($N_c$ in units of $10^6$ cells ml$^{-1}$). Vertical error bars represent the uncertainty on the fits in **a**. Horizontal error bars are the standard deviations of cell concentrations. (**c**) Effective microparticle diffusivity ($D_{eff} - D_0$) as a function of $N_c$ for the three experiments. Diffusivities are rescaled by the ratio of the average CR speeds $\langle v \rangle_S / \langle v \rangle_j$, where $j$ stands for sedimentation (S), spreading (CS) or tracking (HS) experiments. Solid lines are best linear fits to the data. Orange squares: sedimentation experiment (slope $\alpha_S = 1.71 \pm 0.14$ (μm$^2$ s$^{-1}$)/($10^6$ cells ml$^{-1}$); all other values in the same units); green triangles: spreading experiment ($\alpha_{CS} = 1.62 \pm 0.14$); blue circles: Hele–Shaw experiment ($\alpha_{HS} = 1.67 \pm 0.13$). Orange circles and dashed line: direct tracking in the sedimentation experiment ($\alpha_T = 0.074 \pm 0.014$). Black dashed line: fit to the experimental diffusivity obtained by direct tracking in ref. 26 ($\alpha_L = 0.041$). See Supplementary Fig. 5 for a close-up on the low-concentration values. Vertical error bars represent the uncertainty on the fits to obtain the effective diffusivities. Horizontal error bars are the s.d. of the cell concentrations.

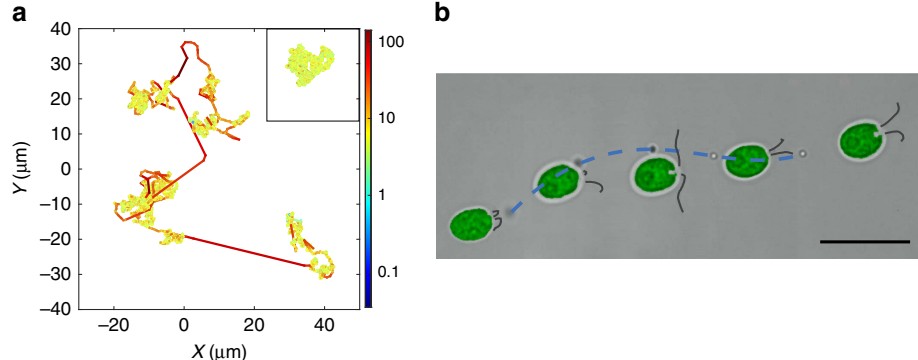

**Figure 2 | Microparticle behaviour within the active suspension.** (**a**) Typical microparticle trajectory ($\sim$210 s) in the Hele–Shaw experiment at $N_c = 4.84 \pm 0.13 \times 10^6$ cells ml$^{-1}$. Colour represents instantaneous speed (colour bar unit: μm s$^{-1}$). The trajectory shows three types of dynamics: Brownian motion and loop-like perturbations (yellow-green blobs) followed by rare and large jumps (red lines). Inset: representative trajectory of a purely Brownian particle in the same setup, lasting $\sim$210 s. (**b**) A representative entrainment event: as the cell swims from the left to the right of the panel, it drives the colloid along the dashed line. Scale bar, 20 μm. The event lasts 1.16 s.

microscopically from direct short-duration tracking of microparticles' trajectories in bulk (mean tracks duration $\delta t_{tracks} = 2.6$ s), as previously done in ref. 26. These measurements return a different estimate, $D_{eff} = D_0 + \alpha_T N_c$ (Fig. 1c, orange circles), with a slope $\alpha_T = 0.074 \pm 0.014$ in reasonable agreement with previous results ($\alpha_L = 0.041$ in ref. 26) but more than 40-times smaller than the sedimentation value $\alpha_S$. The surprisingly large value of $\alpha_S$, larger than any previously reported bulk value, calls for an independent verification. It was tested here at the macroscopic level by following the diffusive spreading of a uniform band of density-matched colloids within a microfluidic device filled with a uniform concentration of cells (for experimental details see the Methods section, Supplementary Fig. 5 and Supplementary Methods 2). The band's profile, initially tight around the middle third of a 2-mm-wide, 60-μm-thick channel and running along its full length ($\sim$10 mm), spreads with a characteristically diffusive dynamics which enables to measure directly the effective diffusivity $D_{eff}$ at different $N_c$ values (Supplementary Fig. 5 and Supplementary Methods 2). The results are shown in Fig. 1c (green triangles) after being

multiplied by the ratio $\langle v \rangle_S / \langle v \rangle_{CS}$ of the cells' swimming speeds to account for their slower motion in the density-matched medium. As before, $D_{eff}$ depends linearly on cell concentration, with a slope $\alpha_{CS} = 1.62 \pm 0.14$ in remarkable agreement with the sedimentation value.

**Microscopic entrainment and diffusion.** The quantitative agreement between steady-state and time-dependent macroscopic measurements suggests that our understanding of the microscopic interaction between particles and microorganisms, based on the effect of the swimmers' far-field flows[29,37–39] and leading to $\alpha_T(\ll \alpha_S)$, is missing a key element. The crucial microscopic insight is provided by long-time tracking of the particles at a range of $N_c$ values. This is achieved here by confining the system within a 26 μm-thick Hele–Shaw cell, which allows us to follow individual colloids for $\sim$200 s. Figure 2a and Supplementary Movie 1 show a typical colloidal trajectory in this confined geometry. The dynamics, which leads to a dramatically larger spreading than simple Brownian motion (Fig. 2a inset), results

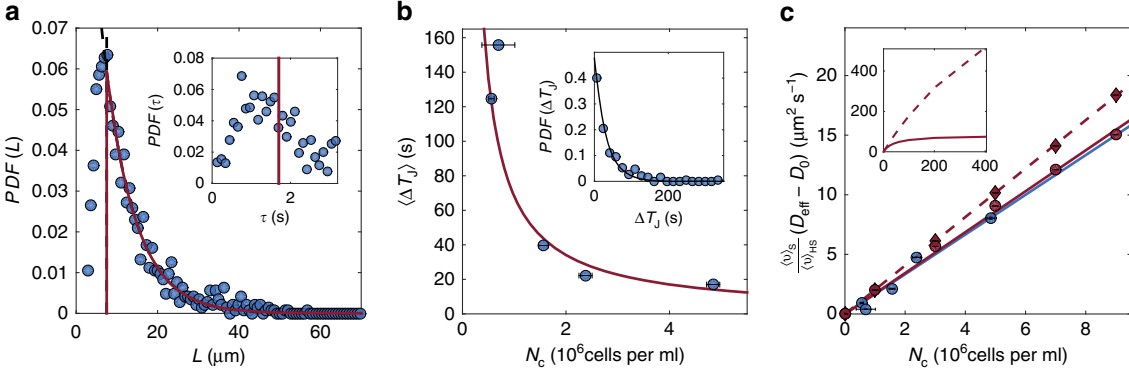

**Figure 3 | Microscopic characterization of particle dynamics.** (**a**) Probability distribution function (PDF) of the end-to-end length $L$ of the jumps. Above $L_T = 7.5\,\mu$m, the distribution is well fitted by an exponential function with characteristic length $L_J = 7.5\,\mu$m (solid red line). Inset: PDF of the duration $\tau$ of the jumps. The average is $\tau = 1.7$ s when considering only jumps of length $L \geq L_T$ (solid red line). (**b**) Mean time interval between consecutive jumps, $\langle \Delta T_J \rangle$, as a function of $N_c$. The red solid line is the hyperbolic fit used in the simulations, $\langle \Delta T_J \rangle = ((68.2 \pm 8)/N_c)$s ($N_c$ in units of $10^6$ cells ml$^{-1}$). Horizontal error bars are the standard deviations of cell concentrations. Inset: PDF of the time interval $\Delta T_J$ between consecutive jumps at $N_c = (1.56 \pm 0.10) \times 10^6$ cells ml$^{-1}$. Black solid line: exponential fit with characteristic time $(31.9 \pm 4)$s. Note that these distributions provide a biased measure of the mean waiting time $\langle \Delta T_J \rangle$, which should be estimated instead from the average number of jump events as discussed in Supplementary Methods 4. (**c**) Effective diffusivities, $D_{eff} - D_0$, from the Hele–Shaw experiment (blue circles) and from the simulations: red circles/solid red line for $\tau = 1.7$ s (slope $1.03 \times \alpha_{HS}$); red diamonds/dashed red line for $\tau = 0.1$ s (slope $1.22 \times \alpha_{HS}$). Vertical error bars represent the uncertainty on the fits to obtain the effective diffusivities. Horizontal error bars are the standard deviations of the cell concentrations. Inset: continuation of the simulated $D_{eff} - D_0$ curves to very high cell concentrations shows saturation to a $\tau$-dependent value.

from the combination of three different effects of well-separated magnitudes. The weakest component is standard Brownian motion, which dominates the dynamics when the algae are more than $\sim 25\,\mu$m (ref. 26) away from the colloid. At closer separation, but before close contact, the far-field flows of the microorganisms induce loop-like trajectories[21,22,31,39,40]. Originally reported in ref. 26, these loops provide the concentration-dependent contribution to the particles' diffusivity previously estimated as $\alpha_T N_c$ (ref. 26). Finally, particles within the near field can be occasionally entrained by the algae over distances up to tens of microns (see Fig. 2b), giving rise to the sudden jumps in the trajectory highlighted as red solid lines in Fig. 2a. These jumps, a fundamental feature of the dynamics never observed previously, are the microscopic origin of the term $\alpha_S N_c$, the unexpectedly large contribution to the particle diffusivity observed in the macroscopic experiments. This is confirmed directly on Supplementary Movie 2, where we show such an entrainment event taking place during a sedimentation experiment. For our 1 $\mu$m-diameter tracers, the jump length $L$ is (mostly) exponentially distributed with a characteristic length $L_J = 7.5 \pm 0.5\,\mu$m, above a threshold value $L_T = 7.5 \pm 0.5\,\mu$m (Fig. 3a) (see Supplementary Methods 3 and Supplementary Fig. 6 for details on the extraction of the jumps from the trajectories). The average displacement is $\langle L \rangle = 12.6 \pm 0.5\,\mu$m, although we did record values of $L$ up to $\sim 70\,\mu$m. The effect of jumps is also clear in the full probability distribution functions of tracer displacements, Fig. 4a,b, which display exponential-like tails much larger than those characteristic of loop-like perturbations only[26]. Both distributions eventually converge to Gaussians -with different variances- as the observation time increases, a consequence of the Central Limit Theorem already predicted in refs 29,31, see Fig. 4 and Supplementary Note 2.

Jumps are fast (Fig. 3a, inset), lasting on average $\tau = 1.5$ s ($\tau = 1.7$ s if considering only jumps with $L \geq L_T$) but they are rare: as shown in Fig. 2b and Supplementary Movie 3, the algae need to meet a microparticle almost head on in order to entrain it ($\sim 65\%$ of all jumps have initial impact parameter $\leq 2\,\mu$m). As a consequence, even at the highest cell concentration probed, $N_C \simeq 5 \times 10^6$ cells ml$^{-1}$, the average time between two consecutive jumps is rather long, $\langle \Delta T_J \rangle \gtrsim 16$ s. Because $\langle \Delta T_J \rangle$ is

dictated by the cell–microparticle encounter rate, for a purely random process one would expect $\langle \Delta T_J \rangle \propto 1/N_c$: Fig. 3b shows that this hypothesis is clearly supported experimentally. Further support comes from the analysis of individual inter-jump intervals, which appear to be exponentially distributed as predicted for a simple Poisson process (Fig. 3b inset, see Supplementary Methods 4 for details on the computation of $\langle \Delta T_J \rangle$).

Computing the mean-square displacement from these long colloidal trajectories offers a microscopic estimate of the effective particle diffusivity $D_{eff}$, Supplementary Fig. 7 and Supplementary Note 3. The results are shown in Fig. 1c (blue circles) after multiplication by the swimming speeds ratio $\langle v \rangle_S / \langle v \rangle_{HS}$ to account for the trivial dependence of the encounter frequency on microalgal speed: $D_{eff}$ is linear with cell concentration, with a slope $\alpha_{HS} = 1.67 \pm 0.13$ in excellent quantitative agreement with both macroscopic values. This agreement confirms that the properties of the colloids' microscopic dynamics, which includes important but rare jumps, have indeed been probed appropriately.

**Numerical simulations.** The agreement between the macroscopic and microscopic diffusivity measurements highlights the importance of entrainment events, and suggests a combination of jumps and (far-field-enhanced) diffusion as a plausible minimal model of microparticles' dynamics, at least over the medium-to-long timescales we are interested in. Consequently, we model the two-dimensional projection of the stochastic trajectory of a colloid in the Hele–Shaw experiments, $(X(t), Y(t))$, as

$$\begin{aligned} dX(t) &= \sqrt{2D_{WJ}}\, dW_X(t) + L\cos(\theta)\, dP(t) \\ dY(t) &= \sqrt{2D_{WJ}}\, dW_Y(t) + L\sin(\theta)\, dP(t) \end{aligned} \quad (1)$$

This combines standard Wiener processes $dW_{X,Y}(t)$ leading to diffusion with diffusivity $D_{WJ}$, with a Poisson process for the jumps, $dP(t)$, characterized by the average time interval $\langle \Delta T_J(N_c) \rangle$ between jumps (Fig. 3b, solid red line). Far-field effects are included at a coarse-grained level by choosing $D_{WJ} = D_0 + \alpha_{WJ} N_c$, where $\alpha_{WJ} = 0.33 \pm 0.05$. This is the effective diffusivity, rescaled by $\langle v \rangle_S / \langle v \rangle_{HS}$, which is measured from the microscopic

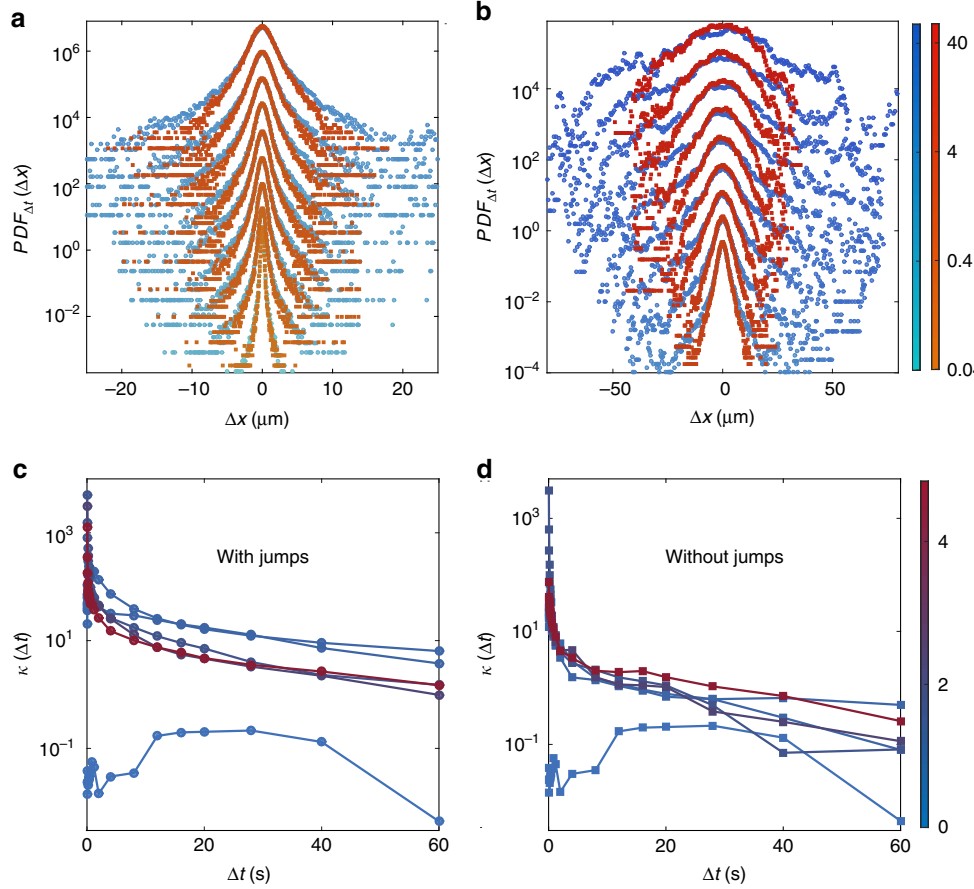

**Figure 4 | Probability distribution functions of particle displacements in the Hele–Shaw experiments.** (**a,b**) Evolution of the PDF of displacements $PDF_{\Delta t}(\Delta x)$ of the colloids with the time interval $\Delta t$ when considering the jumps (blue curves) or not (red curves) for $N_c = (4.84 \pm 0.13) \times 10^6$ cells ml$^{-1}$. The curves have been shifted and separated into two figures for clarity. **a** corresponds to $\Delta t \in [0.04, 0.8]$s, while **b** corresponds to $\Delta t \in [1.2, 48]$s (colour bars: $\Delta t$ in seconds). Both distributions exhibit exponential-like tails at short time which reflect the presence of loop-like perturbations and jumps in the dynamics. (**c,d**) Modified kurtosis $\kappa$ of the PDF of displacements as a function of the time interval $\Delta t$ with/without jumps respectively. The colours code for the cell concentration (unit: $10^6$ cells ml$^{-1}$). The PDFs with jumps show slower convergence to the Gaussian limit, as the plots in **a,b** also suggest.

experimental tracks when the entrainment events have been removed. Notice that $\alpha_{WJ} > \alpha_T$ due to our conservative choice for what constitutes a jump, which in the simulation we draw from an exponential fit to the part of the experimental distribution strictly above $L_T$ (Fig. 3a, solid red line). As a result, the compounded effect of shorter jumps is included within the coefficient $\alpha_{WJ}$. The jumps' orientation $\theta$ is uniformly distributed to give an isotropic process, and their duration $\tau = 1.7$ s is constant.

This dynamics, simulated with a simple acceptance-rejection method (see Methods section), produces trajectories very similar to the experimental, Supplementary Fig. 8 and Supplementary Note 4. The motion is characterized by an effective diffusivity in excellent agreement with the values from the microscopic experiments (see Fig. 3c, red and blue circles respectively), and $\gtrsim 5 \times$ larger than $D_{WJ}$ at the corresponding cell concentration. The simulations, then, highlight the striking influence that jumps have on particle dynamics, despite their rarity. At the same time, they allow us to explore easily parameter values that have not yet been probed experimentally. For example, Fig. 3c (red diamonds and dashed red line) shows that within the experimental range of cell concentrations, even a drastic reduction of the entrainment duration $\tau$ to 0.1 s has only a minimal effect on particle diffusivity. The exact value of $\tau$, however, will have a major influence as soon as $\langle \Delta T_J(N_c) \rangle \sim \tau$, leading to a plateau in the effective diffusivity as the cell concentration grows above a threshold $N_c^*(\tau) \propto 1/\tau$

(Fig. 3c inset), at least as long as collective effects do not modify our single-particle picture.

**Analytical theory.** The experimental and simulation results can be tied together further through a simple continuum theory for the dynamics of microparticles, which rationalizes the dependence of the effective diffusivity $D_{eff}(N_c)$ on the concentration of algae. Here we extend to two dimensions the one-dimensional approach developed in ref. 41. Briefly, we consider two populations whose densities at position $(x, y)$ and time $t$ are $\rho_d(x, y, t)$ and $\rho_b(x, y, t, \phi)$, corresponding respectively to particles diffusing with diffusivity $D_{WJ}$, and to particles moving ballistically with a constant velocity $u$ in the direction $\phi$. Particles switch from diffusion to directed motion and vice versa with constant transition rates $\lambda_d (= 1/\langle \Delta T_J \rangle)$ and $\lambda_b (= 1/\tau)$ respectively. The system then obeys the following set of equations:

$$\frac{\partial \rho_d}{\partial t} = D_{WJ} \Delta \rho_d - \lambda_d \rho_d + \lambda_b \int_0^{2\pi} \rho_b \, d\phi$$

$$\frac{\partial \rho_b}{\partial t} = -u \cos(\phi) \frac{\partial \rho_b}{\partial x} - u \sin(\phi) \frac{\partial \rho_b}{\partial y} + \frac{\lambda_d}{2\pi} \rho_d - \lambda_b \rho_b \qquad (2)$$

which can be easily solved by Fourier–Laplace transform, Supplementary Note 5, yielding the time evolution of the particles' mean-square displacement and hence their diffusivity.

At long timescales this is given by

$$
\begin{aligned}
D_{\text{eff}} &= \frac{D_{\text{WJ}}\lambda_{\text{b}}}{\lambda_{\text{d}}+\lambda_{\text{b}}} + \frac{\lambda_{\text{d}}u^2}{2\lambda_{\text{b}}(\lambda_{\text{d}}+\lambda_{\text{b}})} \\
&= \frac{D_0 + \alpha_{\text{WJ}}N_{\text{c}} + \frac{\langle L\rangle^2}{2}\gamma\nu N_{\text{c}}}{1 + \frac{\langle L\rangle}{u}\gamma\nu N_{\text{c}}},
\end{aligned} \tag{3}
$$

where we wrote $D_{\text{WJ}}=D_0+\alpha_{\text{WJ}}N_{\text{c}}$, $\lambda_{\text{b}}=u/\langle L\rangle$ and $\lambda_{\text{d}}=\gamma\nu N_{\text{c}}$ as the frequency of entrainment events is proportional to the product of the speed $\nu$ and concentration $N_{\text{c}}$ of microorganisms, also called 'active flux'[16]. The quantity $\gamma$ can be interpreted as a cross-section for entrainment: as an alga swims past, the tracer will be entrained -with a given probability- if it is contained within a region of area $\gamma$ around the line of motion of the alga[19,21]. For our Hele–Shaw experiments, using $\nu=\langle\nu\rangle_{\text{HS}}$ and the experimental values of $\langle\Delta T_{\text{J}}\rangle$ (Fig. 3b) we obtain $\gamma=299\pm35\,\text{mm}^2$ (respectively $\gamma=299\pm35\,\mu\text{m}^2$ if $N_{\text{c}}$ is expressed in units of cells ml$^{-1}$ rather than $10^6$cells ml$^{-1}$). At low concentrations, where $\lambda_{\text{d}}\ll\lambda_{\text{b}}$ (or equally $\langle\Delta T_{\text{J}}\rangle\gg\tau$), $D_{\text{eff}}$ becomes

$$
D_{\text{eff}} = D_0 + \alpha_{\text{WJ}}N_{\text{c}} + \frac{\langle L\rangle^2}{2}\gamma\nu N_{\text{c}}, \tag{4}
$$

which is independent of the jumps' duration $\tau$, here equivalent to independence on $u$, as suggested by the simulations. Equation (4) recovers the clear division between thermal, far-field and entrainment contributions to the diffusivity that we previously discussed in the context of microscopic experiments. Notice that the contribution from the jumps, by far the most important in our experiments, is simply what should be expected if we interpreted the colloidal trajectory as a freely jointed chain where bonds with exponentially distributed length of mean $\langle L\rangle$ are added at a rate $\lambda_{\text{d}}=\gamma\nu N_{\text{c}}$.

The coefficient $\alpha_{\text{WJ}}$, representing far-field effects, is proportional to the speed $\nu$ of the microalgae[21,26,29], leading to an overall contribution $(D_{\text{eff}}-D_0)$ which scales with the active flux $\nu N_{\text{c}}$, as originally predicted in ref. 19. Figure 1c shows indeed that the diffusivity curves from different experiments collapse when rescaled by the corresponding velocities. In turn, then, the distribution of jump lengths should be independent of the average velocity of the microorganisms, as expected at low Reynolds numbers. Within the model, however, this proportionality is limited to sufficiently low cell concentrations. As $N_{\text{c}}$ increases and $\langle\Delta T_{\text{J}}\rangle$ becomes closer to $\tau$, nonlinearities become important, and eventually $D_{\text{eff}}$ plateaus to a $\tau$-dependent value as seen in the simulations (Fig. 3c, inset) and well captured by the present model, Supplementary Fig. 9 and Supplementary Note 5.

Finally, the short timescale limit of the particles' mean-square displacement returns $D_{\text{eff}}=x_{\text{d}}D_{\text{WJ}}$, where $x_{\text{d}}$ is the fraction of particles that are in the diffusing state. Estimating $x_{\text{d}}\simeq\langle\Delta T_{\text{J}}\rangle/(\langle\Delta T_{\text{J}}\rangle+\tau)$, we can assume $x_{\text{d}}\simeq1$ within the whole range of cell concentrations probed experimentally. Short timescale tracking of microparticles, then, will inevitably return $D_{\text{eff}}\simeq D_{\text{WJ}}$ rather than the full expression in equation (4). This is the reason for the small diffusivity reported in ref. 26, which we also observe from direct short-duration tracking of microparticles in the sedimentation experiments.

## Discussion

Proposed theoretically either as a consequence of microparticle capture within 'wake bubbles'[21] or as a consequence of Darwin drift[28], particle entrainment by microorganisms was expected to provide at best a contribution similar to that observed for to far-field loops[42] and most likely much smaller[22]. At the same time, lack of experimental evidence for entrainment questioned not just its importance for particle–microswimmer interactions, but its existence as well.

Here we show experimentally not only that entrainment of microparticles by microorganisms exists but also that these rare but large events can dominate particle dynamics, leading in the present case to a diffusivity more than $40\times$ larger than previously reported. Simulation and analytical results support a jump-diffusion process as a good minimal model for the medium-to-long timescales dynamics of the colloids. The simplified continuum model we discuss provides a theoretical support for the observed dependence of the experimental diffusivities on the active flux $\nu N_{\text{c}}$, already introduced in the bacterial context[16,17,43]. At the same time, it clarifies that long-duration particle tracking is necessary to sample correctly the microscopic dynamics and recover the real long timescale impact of microorganisms.

Although we cannot yet pinpoint the specific mechanism leading to entrainment, the structure of the near-field flow is likely to play a crucial role. The entrainment we observe requires almost head-on collisions, which is likely to be facilitated by the type of stagnation point found (on average) in front of the cell[37,38]. After reaching the cell apex, the entrained particles slide down the sides of the cell body along a high-shear region almost co-moving with the microorganism, and are eventually left behind having spent slightly more than half of the jump in the front part of the cell ($54\pm9\%$). The no-slip boundary on the bodies of microorganisms, and the stagnation points in their flow fields, have in fact been argued to play a major role in large microparticle displacements[5,21,29], which are seen here to dominate the effective diffusivity. Our results support these conjectures.

Eukaryotic microswimmers with multiple front-mounted flagella will share much of the near-field flow structure found in *Chlamydomonas*: entrainment is then likely to be a generic feature of this whole class of microorganisms. Several of these species are predators[2,3], and prey on cells of size similar to our plastic particles. Front-mounted flagella, then, would spontaneously lead to contact with the prey at a predictable location on the cell body within easy reach of the flagella, and therefore facilitate the ingestion of both natural preys and environmental microplastics.

## Methods

**Cell culture.** Cultures of CR strain CC125 were grown axenically in a Tris-Acetate-Phosphate medium[32] at 21 °C under continuous fluorescent illumination (100 $\mu$E m$^{-2}$ s$^{-1}$, OSRAM Fluora). Cells were harvested at $\sim 5\times10^6$cells ml$^{-1}$ in the exponentially growing phase, then centrifuged at 800 r.p.m. for 10 min and the supernatants replaced by DI-water (sedimentation experiment) or by a Percoll solution (tracking and spreading experiments; Percoll Plus, Sigma) already containing the desired concentration of PS colloids (Polybead Microspheres, diameter $d=1\pm0.02\,\mu$m).

**Microfluidics and microscopy.** The Percoll solution (38.5% vol/vol) was made to density-match the beads for the Hele–Shaw and spreading experiments, while preserving the Newtonian nature of the flows[33]. The appropriate solution was injected into either 185 $\mu$m (sedimentation experiment) or 26 $\mu$m (tracking experiment) thick Polydimethylsiloxane-based microfluidic channels previously passivated with 0.15%w/w BSA solution in water. The microfluidic devices were then sealed using photocurable glue (Norland NOA-68) to prevent evaporation. Regarding the spreading experiment, the band of colloids was initiated in a three-arms fork-shape channel (60 $\mu$m thick) by injecting at the same flow-rate (using a PHD 2,000 Harvard Apparatus syringe pump and high-precision Hamilton 50 and 100 $\mu$l gas-tight syringes) the CR + beads Percoll solution in the central arm and the CR Percoll solution only in the two sided arms. Colloids were observed under either bright-field (sedimentation experiment) or phase contrast (tracking and spreading experiments) illumination on a Nikon TE2000-U inverted microscope. A long-pass filter (cutoff wavelength 765 nm) was added to the optical path to prevent phototactic response of the cells. Stacks of 200 images at $60\times$ magnification were acquired at 10 f.p.s. (camera Pike F-100B, AVT) layer by layer by manually moving the plane of focus in order to reconstruct the density profiles for the sedimentation experiment. We used a $\times40$ oil immersion objective (Nikon CFI S Fluor $\times40$ oil) combined with an extra $\times1.5$ optovar magnification. The condenser iris was completely opened to minimize the depth of field. Regarding the Hele–Shaw experiment, several movies of $2\times10^4$ images were recorded at 25 f.p.s. (camera Pike F-100B, AVT) using a $20\times$ phase contrast objective (Nikon LWD ADL $20\times$ F). Particles trajectories were then digitized using a standard Matlab particle

tracking algorithm (The code can be downloaded at http://people.umass.edu/kilfoil/downloads.html). Finally the spreading dynamics of the band of colloids was probed by recording the system for a few hours at ×10 magnification (Nikon ADL 10×I) and low frame rate (0.5 f.p.s.) using a Nikon D5000 DSLR camera. The colloids were then featured using the same Matlab particle tracking algorithm in order to reconstruct the density profiles. In all experiments, the concentration of algae was determined *in situ* by imaging the system under dark-field illumination at low magnification (objective Nikon CFI Plan Achromat UW 2×).

**Numerical Simulations.** The parameters of the simulation correspond to the red lines in Fig. 3a,b. For each algae concentration, 1,000 trajectories of 2,000 s have been simulated using a time-step $\delta t = 0.004$ s, which is 10 times smaller than the acquisition period in the Hele–Shaw experiment. At each time step a random walk with diffusivity $D_{WJ}$ is performed. To simulate the Poisson process, we draw at each time step a random number in the open interval $]0;1[$. If this number is within the centred closed interval $[(1 - \delta t / \langle \Delta T_J \rangle)/2; (1 + \delta t / \langle \Delta T_J \rangle)/2]$, then we perform a jump that lasts $\tau = 1.7$ s (or 0.1 s) with a length taken out of the distribution $PDF(L)$, along the direction corresponding to $\theta$, which is uniformly distributed in $[-\pi, \pi]$. This technique allows to simulate accurately the Poisson process[44]. We stress that within the simulation, once the particle has entered a jump, it will escape it after exactly $\tau = 1.7$ s (or 0.1 s). This is done in order to approximate the experiments, which show that the distribution function of jumps' duration is tighter around the mean than that of jumps' lengths. Notice that during the jumps the random walk component is switched off. After the jump has been performed, a new random number is picked in order to choose between a random step or a jump and continue the jump-diffusion process.

**Data availability.** The authors declare that the data supporting the findings of this study are available on request.

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

## Author contributions

R.J., D.O.P., V.K., M.P. designed the study; R.J. performed the experiments, analytical and numerical modelling; R.J. and M.P. analysed the results; R.J., V.K., M.P. wrote the manuscript.

## Additional information

**Competing financial interests:** The authors declare no competing financial interests.

