## [Peer Review File · Nature Communications]

Reviewers' comments:

Reviewer #1 (Remarks to the Author):

The manuscript of R. Jaenneret et al. presents an experimental and theoretical study of the 'activity-induced' diffusion of micro-sized objects in suspensions of eukaryotic cells. This subject is of large interest for biological and ecological applications. It is also present a challenge for the modern theories of non-equilibrium statistical mechanics.

The main finding of the paper is experimental and based on the idea to study sedimentation profiles of micron-sized particles. Albeit simple, this idea allows to probe into the dynamics of activity-induced diffusion on very long time scales. Although this regime is particularly relevant for applications and non-equilibrium theories, it has not been studied experimentally previously.

Studies of activity-induced diffusion focus on three major questions: a) whether the mixing process is diffusive; b) what is the magnitude of the effective diffusion coefficient; and c) what is the form of the particle displacement distribution.

In the current study the authors establish that for suspensions of eukaryotic cells that are larger than the size of microscopic particles: a) the particle mixing process is diffusive as evidenced by the exponential sedimentation profiles; b) the effective diffusion is at least an order of magnitude higher than has been believed based on previous experimental [Leptos et al, PRL 2009], theoretical [Lin et al JFM 2011, Pushkin et al PRL 2013; Thiffeault PRE 2015] and numerical studies [Underhill, PRL 2008, Ischikawa PRE 2010; Lin et al JFM 2011, Pushkin et al J Stat Mech 2014; Morozov et al Soft Matter 2014; Thiffeault PRE 2015]; c) the long-time particle displacement distributions have exponential tails.

The mechanisms leading to the diffusive character of mixing in dilute suspensions of swimmers have been first discussed by Thiffeault and Childress, PRA 2010; Lin et al JFM 2011; Zaid et al J Roy Soc Int 2011; Pushkin et al PRL 2013. In particular, the idea to distinguish the entrainment by the swimmers in the near-field and the contributions of loop-like motion of particles induced by numerous far-away swimmers as separate and different mechanisms leading to the diffusive behaviour was proposed in Pushkin PRL 2013. (See the discussion why both mechanisms lead to a diffusive process in the recent paper of Thiffeault PRE 2015. The term 'entrainment' was first discussed and used in the current context by Pushkin et al JFM 2013.).

The current work of Jaenneret et al. allows establishing unequivocally that the activity-induced diffusion in suspensions of eukaryotic cells is due to the entrainment mechanism and that the fat exponential tails of the particle displacement distribution dominate the process.

These findings are new, dramatic, and important. They appear to be well-supported by additional experiments performed by the authors and detailed in the supplementary material. The theoretical model presented by the authors is adequate in describing the ensuing diffusive process and helps to understand why the large values of the diffusion coefficient have not been observed in previous short time-scale measurements.

What the current work does not explain is the mechanistic origin of the particle capture mechanism, but elucidating it may be left for the future work.

The main drawbacks of this paper are poorly written introduction and discussion. The introduction refers to a number of rather arbitrary papers that are flashy and new but not necessarily important

and valid, particularly in the part related to the effective temperature approach. The essential references here are missing. Also, the discussion of the past work on biomixing is somewhat arbitrary.

Also, the discussion of the particle diffusion kinetics is tucked away in the analytical theory part of the paper. The basic expression for the diffusion coefficient due to the entrainment kinetics (first discussed by Underhill PRL 2008 and Lin et al 2011) is not discussed properly by the present authors: $D \sim \gamma L^2 V N_c$, where γ is the collisions cross-section area and L is the average entrainment length. Note that γ remains undefined in the current manuscript but its numerical value comes out $\gamma \approx 300 \text{ mm}^2$. This is an enormously high value at odds with the assertion that only near-collisions contribute to the long particle displacements. ($\gamma = \pi * b^2$, where b is the impact parameter that, according to the authors, $b \leq 2 \text{ microns}$). The authors should discuss this discrepancy as it is at the heart of the large values of the observed diffusion coefficient.

In the discussion it is essential to refer to the paper by Lin et al JFM 2011, who first pointed out that the exponential tails may be produced due to the particle entrainment (they did not use this term though) by the stagnation point in front of the swimmer. They also noted that these tails become apparent only after a very long time. A word of caution: these 'long-time' exponential tails have a different nature than the 'short-time' exponential tails reported by Leptos et al. The latter seem to be due to the scarcity of close collisions experienced by tracer particles during the short experimental times, which leads to the central limit theorem breakdown.

In summary, I believe that this manuscript presents an important work likely to impact strongly this field of science and that it should be published in Nature Communications after a revision.

Reviewer #2 (Remarks to the Author):

The manuscript "Entrainment dominates the interaction of microalgae with micron-sized objects" by Jeanneret, Kantsler & Polin investigates particle dynamics in suspensions of the green algae *Chlamydomonas reinhardtii*. The authors measure particle diffusivities in suspensions of varying algae concentration through direct microscopic tracking, sedimentation, and micro-channel spreading experiments. The authors find particle diffusivities higher than previously reported in the literature and argue that the effect is due to rare entrainments of the particles by the swimming algae, which lead to large displacements. The authors present simulations and theory in agreement with their experimental results.

The manuscript provides important insights into the transport of micron-sized particles by *Chlamydomonas reinhardtii*, and their results are appealing to a wide range of audiences, such as that of Nature Communications. The manuscript also contains a wealth of new data. However, there are a number of concerns regarding the sedimentation and spreading experiments, which must be addressed before the manuscript can be recommended for publication. Below are some of my concerns:

Major concerns:

- 1- The authors have not mentioned that *Chlamydomonas reinhardtii* is gyrotactic. This commonly leads to plumes and bioconvection. Are the authors certain that have no large-scale flow in the sedimentation experiments? If not, why have they not developed? Bioconvection and large-scale flows could lead to possibly much higher particle diffusivities that are not comparable to the particle tracking diffusivities.

2- The authors should comment more on the role of viscosity in the experiments. In particular,

2a. The viscosity of the Percoll solutions is not provided.

2b. The viscosity is important because it affects the particle's Brownian diffusion and ultimately Peclet number $Pe = UL/D$, where U , L , and D are the microorganisms swimming speed, microorganisms length, and the particles Brownian diffusion coefficient. The Peclet number influences the particle's ability to be actively transported by the microorganism [1]. It is useful to provide the Peclet number for more general comparisons.

3. The algae density variation with height (Fig. 4) is quite significant (factor of 2) for the two highest concentrations shown. Ideally, the concentration of algae should be constant throughout. The authors should note this in the main-text of the manuscript.

4. The authors claim the sedimentation profiles are steady state. Are the authors sure it is at steady state? How long does it take the system to reach a steady state? Are the algae swimming kinematics unchanged during the duration of experiment? These are important checks that can be included in the SI.

5. The authors should relate their findings to the large tails and higher probabilities of large displacements perhaps reported elsewhere. The main critical point is that they follow the particle trajectories in the tracking experiments for very long times 210 seconds or so and are able to capture these rare events. So it makes sense to comment on if others see signatures in the pdf of displacements when particles are tracked for as long in similar papers on CR. They see a mean end to end jump length of 7.5 microns with some jumps of 70 microns which seems unbelievably long! Probably as a consequence of head-on encounters resulting in strong and sustained entrainment.

6. I also would like a little more discussion of the loops and straight long trajectories are reflected in measurements of displacement pdf's from other studies (Figure 3(a) from other papers for instance).

7. The authors may want to justify for general readers why they scale diffusivities with $\langle v \rangle_s / \langle v \rangle_{2D}$ in reporting α_{2D} .

Minor concerns:

1. It may be useful for the authors to include a table (perhaps in Fig. 1) that provides the measured slopes for the 3 different experiments to ease comparison.

2. The authors claim the high diffusivities are due to rare collision events. Are any of these observed in the sedimentation or spreading experiments?

References:

[1] Kasyap TV, Koch DL, Wu M. Hydrodynamic tracer diffusion in suspensions of swimming bacteria, Phys. Fluids: 26, 081901 (2014).

Reviewer #3 (Remarks to the Author):

In their paper the authors study further how the swimming behavior of a type of eukaryotic flagellates (i.e. *C. reinhardtii*) induces an enhancement in the diffusion dynamics of microscopic colloidal particles contained in the same solution of the microalgae. Differently from similar phenomena observed in active baths driven by bacteria (i.e. *E. coli*), they explain that this enhanced diffusion is dominated by rare entrainment events that transport the passive colloids over long distances. They then propose a model based on Poisson-distributed jumps to describe the microdynamics of the motion of the colloids.

This work presents an interesting characterization and modeling of the diffusive microscopic dynamics of passive colloids interacting with microalgae. However, in this respect, what matters in the interaction between passive particles and active particles (particularly in a homogenous environment such as that studied by the authors) is the long-term behavior rather than the microscopic details that lead to it. As the authors say in their introduction, the long-term behavior, i.e. the enhancement in diffusion for passive colloids in the same system, was already observed in Refs. [25-26]: Ref. [26] is particularly interesting since it shows the same dynamics of rare long-range transport events (reported by the authors of the present manuscript and, here, named as entrainment) that sometimes can extend over distances of a few tens of microns when the microalgae are passing close enough to the colloids. The same Ref. [26] studies the effect of microalgae concentration on the enhanced diffusion of the colloids and, for example, they report a 900-fold enhancement of diffusivity for a cell concentration of $\approx 1.3 \times 10^8$ cells/mL, perfectly compatible with the fitting equations put forward in the current manuscript, and in contrast to the authors' claim that they observe a diffusivity more than 40 x larger than previously reported.

Coming to the actual mechanism of entrainment itself, this mechanism is not counterintuitive considering that it has already been experimentally reported on similar active systems that turbulence can lead to a significant enhancement in diffusion (see, e.g., Zhang et al., *Europhys Lett* 87, 48011 (2009)). Moreover, the data and the analysis performed by the authors are purely phenomenological and do not offer any deep insight on the actual physical mechanism itself, which would be more interesting if assessed experimentally. Is entrainment due to hydrodynamic coupling as suggested? Or are the algae simply pushing the colloids head-on?

Considering previous literature on the topic, both experimental and theoretical, the main novelty of this work, I believe is the proposal of a relatively different model of micro-dynamics (with Poisson-distributed jumps) to explain the enhanced diffusion of the passive particles in the presence of motile microalgae. In my opinion, this by itself is not novel enough (or of broad interest for specialists in the field) to warrant publication in *Nature Communications*.

Moreover, I have also a few technical remarks:

- Considering the different types of experiments (i.e. sedimentation, spreading, quasi-2D tracking) that were tried and whose data compared, it would really be worth including detailed schematics of the different setups to help visualize them.
- I am a bit puzzled by the authors' choice to express the enhancement in diffusion as $Deff-D0$ rather than $Deff/D0$ (in Fig. 1, for example)
- The authors compare diffusion coefficients between bulk experiments, quasi-2D experiments, and, as far as I can tell, quasi-1D experiments in the spreading case (schematics would help to understand the actual configuration better), but, at least looking at the equations in the supplementary information, they do not seem to take dimensionality into account when comparing the different values of diffusivity. If dimensionality is not taken into account, the different experiments cannot be directly compared unless opportunely normalized.

- Related to the previous point, Ref. [26] reported that a reduced dimensionality leads to a major enhancement of diffusion, while the authors seem to see no such difference. Why is that?
- I wonder why the authors did not also use Percoll in the sedimentation experiment in order to have same viscosity and data directly comparable to the other experiments?
- In the sedimentation experiment, the authors compare the diffusion calculated from the effective gravitational length and that calculated from short-duration tracking of microparticles and find different values for it. First, can they exclude the effect of drift due to gravity in the first measurement? Then, if no drift is at play, this difference could be explained also by the fact that the two measurements probe different timescales. For active particles (or colloids in active baths) the diffusive dynamics typically show three different timescales: at very short time scales, these dynamics are diffusive following standard thermal diffusion, at intermediate time scales, they become superdiffusive (growing quadratically in time), and finally they go back to linear diffusion with an enhanced diffusion coefficient. Considering that the tracking was done at short-duration intervals, couldn't it be that the two sets of measurements are either probing intermediate or long-time scales? If that's the case, then the difference in diffusivity is not surprising.
- In Fig. 3, the authors fit the distribution of jumps with a Poisson distribution. This fitting seems quite good at long jumps; however, it seems to be quite off when looking at short jumps that are also the most probable ones. I wonder if the Poisson distribution is actually the most appropriate distribution to describe the observed microscopic dynamics?

Reviewers' comments:

Reviewer #2 (Remarks to the Author):

The authors have addressed all my concerns and I now recommend the manuscript for publication.

Reviewer #3 (Remarks to the Author):

First of all, I would like to thank the authors for clarifying the difference between their work and reference [26] that was not clear to me in the first iteration. Considering this key difference, I agree with the authors that this finding, if confirmed, deserves in principle to appear in an outlet, such as Nature Communications.

Regarding the comments on the effect of dimensionality on the effective diffusion coefficient (as reported in Ref. [26]), this effect seems to be an intrinsic characteristic of particles that show an enhanced diffusivity. While for Brownian particles the diffusion coefficient does not depend on the dimensionality of the system as the authors correctly point out in their reply, this seems not to be the case for systems with enhanced diffusion, as for example, reported in this theoretical work: M. E. Cates and J. Tailleur, EPL, 2013, 101, 20010. I remain therefore puzzled by the fact that the authors do not observe this effect at all. Can they please comment on this? Generally speaking, I think that adding some explanations about the dependence of the diffusivity on the dimensionality in the manuscript would be helpful for the general reader.

Response to Referee #1

We thank the Referee for her/his very positive and much useful report which helped us improving our manuscript. We have taken all her/his suggestions into account.

The main drawbacks of this paper are poorly written introduction and discussion. The introduction refers to a number of rather arbitrary papers that are flashy and new but not necessarily important and valid, particularly in the part related to the effective temperature approach. The essential references here are missing. Also, the discussion of the past work on biomixing is somewhat arbitrary.

We have now revised both introduction and conclusions to take into account essential references that were previously missing.

Also, the discussion of the particle diffusion kinetics is tucked away in the analytical theory part of the paper. The basic expression for the diffusion coefficient due to the entrainment kinetics (first discussed by Underhill PRL 2008 and Lin et al 2011) is not discussed properly by the present authors : $D \sim \gamma L^2 V N_c$, where γ is the collisions cross-section area and L is the average entrainment length. Note that γ remains undefined in the current manuscript but its numerical value comes out $\gamma \approx 300 mm^2$. This is an enormously high value at odds with the assertion that only near-collisions contribute to the long particle displacements. ($\gamma = \pi b^2$, where b is the impact parameter that, according to the authors, $b \leq 2$ microns). The authors should discuss this discrepancy as it is at the heart of the large values of the observed diffusion coefficient.

We thank the Referee for highlighting this point, which indeed needed a clearer discussion. We have now amended the “Analytical theory” section within the main text to be more accurate and complete. The value of the cross-section area γ that we wrote originally was the collision cross-section per 10^6 cells and not per individual cell. We choose this convention to be consistent with the unit of the α 's introduced earlier in the text. The value of γ per cell is then $\gamma = 299 \mu m^2$, which leads to an impact parameter $b \simeq 10 \mu m$. We agree that the former convention might have been confusing : we now provide both values in the main text and a clear explanation of their interpretation. We now also specify in the text that γ has been obtained from our measurement of $\langle \Delta T_J \rangle = 1/\lambda_d$. Estimating this quantity directly from $\alpha_{2D} - \alpha_{WJ}$ leads to $\gamma = 205 \mu m^2$, leading to $b \simeq 8 \mu m$ not far from the other estimates. This number is also coherent with our direct measurements, which show entrainments up to impact parameters $b \sim 10 \mu m$, although the majority of entrainments do happen for $b \lesssim 2 \mu m$.

In the discussion it is essential to refer to the paper by Lin et al JFM 2011, who first pointed out that the exponential tails may be produced due to the particle entrainment (they did not use this term though) by the stagnation point in front of the swimmer. They also noted that these tails become apparent only after a very long time. A word of caution : these 'long-time' exponential tails have a different nature than the 'short-time' exponential tails reported by Leptos et al. The latter seem to be due to the scarcity of close collisions experienced by tracer particles during the short experimental times, which leads to the central limit theorem breakdown.

We followed the Referee's suggestion, and now explicitly refer to the paper by Lin et al JFM 2011 in the discussion on the role of the stagnation point. Moreover, we have added a paragraph in the text regarding the distribution of displacements and the transient tails, which we measure experimentally both including and excluding entrainment events. A figure and more detailed analysis of these distributions have been added in the Supplementary Informations (Fig.7 in SI). In short, we observe that at small time interval Δt , the two distributions (with and without jumps) are very similar

and include exponential-like fat tails indicative of the temporal heterogeneity of the dynamics. The difference between the two distributions increases with Δt , highlighting the importance of entrainment events : the tails become bigger when considering the jumps. Finally, following the evolution of the kurtosis we see that as Δt is increased both distributions converge to Gaussians, albeit with different standard deviations. This limit, already predicted in Thiffeault PRE 2015, is expected as a consequence of the Central Limit Theorem. As expected, the distribution without jumps converges quicker than the other, because entrainments are rarer than others CR-tracer interactions (loops) and lead to longer displacements.

Response to Referee #2

We thank the Referee for her/his encouraging and much useful report We have taken all her/his suggestions into account, and we provide detailed answers to her/his questions and comments below.

Major concerns :

1- The authors have not mentioned that *Chlamydomonas reinhardtii* is gyrotactic. This commonly leads to plumes and bioconvection. Are the authors certain that have no large-scale flow in the sedimentation experiments? If not, why have they not developed? Bioconvection and large-scale flows could lead to possibly much higher particle diffusivities that are not comparable to the particle tracking diffusivities.

It is true that CR is bottom-heavy and has therefore a tendency to swim upward in a gravitational field. Although this behaviour can lead to bioconvective flows, we did not observe any evidence of bioconvection in our sedimentation experiments. We now explicitly mention this in the paper.

As the Referee will undoubtedly know, bioconvection appears only above a critical cell concentration, for a given layer depth (i.e. channel height), or *vice versa* above a critical layer depth for a given cell concentration. To the best of our knowledge these critical quantities are not known for CR. However, Hill and collaborators have shown that in the related species *Chlamydomonas nivalis* (which is similar in shape, size and swimming behaviour to CR) the critical layer depth is $\simeq 1$ cm at an average concentration of $\simeq 10^6$ cells/ml [1]. This length scale is two orders of magnitude higher than the thickness of the device used for the sedimentation experiment ($\simeq 200 \mu\text{m}$). Given that the maximum concentration used in our sedimentation experiment is just twice the average concentration used in [1] ($\simeq 2.3 \times 10^6$ cells/ml), the absence of bioconvection in our sedimentation experiments is not surprising.

Finally, the absence of bioconvection in the sedimentation experiments is also consistent with the observed agreement between the diffusivities measured in all three experiments, including the quasi-2D geometry. In conclusion, the diffusivities measured in our sedimentation experiment are only due to the direct interaction of individual swimmers with the tracers.

2- The authors should comment more on the role of viscosity in the experiments. In particular,
2a. The viscosity of the Percoll solutions is not provided.
2b. The viscosity is important because it affects the particle's Brownian diffusion and ultimately Peclet number $Pe = UL/D$, where U , L , and D are the microorganisms swimming speed, microorganisms

length, and the particles Brownian diffusion coefficient. The Peclet number influences the particle's ability to be actively transported by the microorganism [1]. It is useful to provide the Peclet number for more general comparisons.

We have followed the Referee's suggestions and now give explicitly the values of the viscosity of the Percoll solution and the Peclet numbers. We measured the Percoll viscosity using a U-tube apparatus and found $\eta_{Percoll} = (1.5 \pm 0.1)\eta_{water}$, consistent with the measured equilibrium diffusivities of the beads in water and in the Percoll solution, $D_{0,water} = 0.40 \pm 0.01 \mu\text{m}^2 \cdot \text{s}^{-1}$ and $D_{0,Percoll} = 0.28 \pm 0.01 \mu\text{m}^2 \cdot \text{s}^{-1}$. The Peclet numbers are of the order of 2000 for all experiments.

3. The algae density variation with height (Fig. 4) is quite significant (factor of 2) for the two highest concentrations shown. Ideally, the concentration of algae should be constant throughout. The authors should note this in the main-text of the manuscript.

We have followed the Referee's suggestion and now explicitly mention in the main text the fact that, in the sedimentation experiment, the algal density is slightly non-uniform with respect to the distance from the bottom of the channel.

4. The authors claim the sedimentation profiles are steady state. Are the authors sure it is at steady state? How long does it take the system to reach a steady state? Are the algae swimming kinematics unchanged during the duration of experiment? These are important checks that can be included in the SI.

In our sedimentation chambers, the time required for a single $1 \mu\text{m}$ -diameter polystyrene colloid to sediment from the top to the bottom surfaces is ~ 1.5 h. Consequently, we have decided to let the system rest for $\simeq 3$ hr before recording the concentration profiles, as a safe waiting time for the equilibration of the sedimentation profile in absence of microorganisms. Our direct measurements show that after this time the tracers' profile is indeed exponential, and the diffusivity derived from the gravitational length ($D_0 = 0.40 \pm 0.01$) is in excellent agreement with that measured from short-time tracking ($D_0 = 0.41 \pm 0.01$). Therefore, our waiting time is sufficient for the system to reach equilibrium without algae. Given that the maximum volume fraction of algae we work with is small ($\lesssim 0.15\%$), the tracers' sedimentation speed with algae will be the same, and therefore we expect that our waiting time will be adequate also in this case. In fact, given that the particles' diffusivity in presence of algae is much larger, the steady state should be reached even quicker than in the equilibrium system. The agreement between the effective diffusivities obtained across all three experiments, which are measured in fundamentally different ways, provides a strong quantitative evidence that the sedimentation profiles were indeed all in steady state. Regarding the algae kinematics, we have measured their speeds in situ before and after actually conducting the experiments for both the quasi-2D and the spreading experiments, and did not notice any alteration of their swimming behaviour.

5. The authors should relate their findings to the large tails and higher probabilities of large displacements perhaps reported elsewhere. The main critical point is that they follow the particle trajectories in the tracking experiments for very long times 210 seconds or so and are able to capture these rare events. So it makes sense to comment on if others see signatures in the pdf of displacements when particles are tracked for as long in similar papers on CR. They see a mean end to end jump length of 7.5 microns with some jumps of 70 microns which seems unbelievably long! Probably as a consequence of head-on encounters resulting in strong and sustained entrainment.

Alas, we are not aware of other studies of CR-enhanced tracers diffusivity that followed the colloids for as long as we did. This probably explains why the entrainment process has not been reported

previously. As a consequence we are not able to compare our results with existing datasets.

Regarding the distribution of jump lengths, we agree that the existence of exceedingly long entrainments can be surprising at first sight. However, they are completely real, and the Referee is right in guessing that they relate to really head-on collisions. The high-speed movie accompanying the paper is a good example of this dramatic effect : this particular entrainment leads to a jump of $\sim 45 \mu\text{m}$.

6. I also would like a little more discussion of the loops and straight long trajectories are reflected in measurements of displacement pdf's from other studies (Figure 3(a) from other papers for instance).

We thank the Referee for this interesting comment, which echoes a similar one from Referee #1. Following this advice, we have added a paragraph in the text and a full section in the Supplementary Informations where we explicitly compare the PDF of tracer displacements at increasing time-lags Δt for the full dynamics, with that obtained excluding entrainment events. In short, we observe that at small Δt , the distributions with and without jumps are very similar and present fat exponential-like tails highlighting the temporal heterogeneity of the dynamics. The difference between the two distributions increases for larger Δt showing the effect of entrainment events : the tails become bigger when considering the jumps. However, the kurtosis of the two distributions indicates that, as Δt increases further, both functions converge to Gaussians, as predicted in Thiffeault PRE 2015 and indeed expected from the Central Limit Theorem. These two Gaussians, however, have different variances. This convergence is faster for the distribution without jumps, which is also expected.

7. The authors may want to justify for general readers why they scale diffusivities with $\langle v \rangle_s / \langle v \rangle_{2D}$ in reporting α_{2D} .

We have added a comment in the text to explain this rescaling.

Minor concerns :

1. It may be useful for the authors to include a table (perhaps in Fig. 1) that provides the measured slopes for the 3 different experiments to ease comparison.

We have now added these values as an inset to Fig. 1c).

2. The authors claim the high diffusivities are due to rare collision events. Are any of these observed in the sedimentation or spreading experiments ?

We have now added a movie to the Supplementary Informations showing a colloid being entrained during a sedimentation experiment. Regarding the spreading experiment, unfortunately we had to use a rather low frame rate (0.5 Hz) to be able to monitor the evolution of the profile for a sufficiently long amount of time. At this frame rate it is not possible to have a direct evidence of single particles being entrained, as individual entrainment events are too fast. However, the rescaling of all the effective diffusivities on a master curve seen in Fig. 1c), and the fact that we do observe directly individual entrainment events in both the sedimentation and quasi-2D experiments, provide a very strong support to the conclusion that the same process is at play in all three experiments.

[1] N.A. Hill, T.J. Pedley, and J.O. Kessler, J. Fluid Mech. **208**, 509-543 (1989)

Response to Referee #3

We thank the Referee for her/his report which helped us improving our manuscript. We have taken all her/his suggestions into account.

This work presents an interesting characterization and modeling of the diffusive microscopic dynamics of passive colloids interacting with microalgae. However, in this respect, what matters in the interaction between passive particles and active particles (particularly in a homogenous environment such as that studied by the authors) is the long-term behavior rather than the microscopic details that lead to it. As the authors say in their introduction, the long-term behavior, i.e. the enhancement in diffusion for passive colloids in the same system, was already observed in Refs. [25-26] : Ref. [26] is particularly interesting since it shows the same dynamics of rare long-range transport events (reported by the authors of the present manuscript and, here, named as entrainment) that sometimes can extend over distances of a few tens of microns when the microalgae are passing close enough to the colloids. The same Ref. [26] studies the effect of microalgae concentration on the enhanced diffusion of the colloids and, for example, they report a 900-fold enhancement of diffusivity for a cell concentration of $\sim 1.3 \times 10^8$ cells/mL, perfectly compatible with the fitting equations put forward in the current manuscript, and in contrast to the authors' claim that they observe a diffusivity more than 40 x larger than previously reported.

The Referee raises two related but conceptually separate points : i) that it is the long-timescale diffusivity enhancement that is important, rather than the actual mechanism leading to it ; ii) that rare long-range transport events identical to the entrainments described here were already reported in Ref [26], which anyway reported a much larger diffusivity enhancement than in our paper.

Let us start by addressing the latter first. It is indeed true that large amplitude displacements were reported in [26]. Although they are not discussed to much extent in the main paper, an explicit example of such an event can be seen in a movie on Dr. Jeffrey Guasto's website¹. The movie shows clearly that the beads are pulled at some distance at the back of the alga. Out of the hundreds of examples of entrainment events we recorded, we have never observed this dynamics. In all our experiments, the microalgae push the entrained beads following direct collisions, as explicitly mentioned in the main paper e.g. in the "Discussion" section. We can conclude that the two phenomena are not the same, and in particular that dragging the tracers from behind is definitely not the mechanism leading to the large diffusivities we observe both in bulk and in our Hele-Shaw cells (quasi-2D experiments). As we comment in more detail in our reply to a later question by the Referee (on the effect of dimensionality, see below), confinement by a soap film has a dramatic effect on the microscopic flows generated by the microalgae, increasing both their magnitude and range (flows decay with distance r from the swimmer as $1/r$ rather than the bulk case of $1/r^2$). This considerable change of flow fields is most likely the reason for the dragging behaviour observed in [26], which appears to be very specific to the 2D film case, and not representative of tracer dynamics either in bulk or under strong confinement by solid surfaces. It is important to notice that while bulk and wall-confined systems are clearly mimicking common naturally occurring situations (bulk : e.g. planktonic microorganisms ; wall-confinement : e.g. microbenthos in the upper sediments of all coastal areas, general soil microorganisms, motile human parasites like Plasmodium), it is rather more difficult to find examples where of the the peculiar results obtained for microorganisms swimming in a soap film are relevant (although we agree that from a purely theoretical perspective this is an interesting system).

The significantly stronger flow fields induced by soap film confinement are also the reason for the very large diffusivities observed in [26]. Again, this is peculiar to the specific type of confinement used in that study, and not generalisable to other geometries. Our results show clearly a very large difference

1. <http://sites.tufts.edu/guastolab/research/biomixing-by-swimming-cells/>

between the diffusivities we measure and those reported previously *for the same* geometry. It is this striking difference that we refer to in the text when comparing to previous results : we have now modified the text in the introduction to make this point clearer. This difference is also what prompted us to do a detailed microscopic investigation which uncovered the existence of entrainments.

This leads us to address the first point, on the importance of the detailed mechanism leading to enhanced diffusivity. Our results show that the work conducted in ref [25] (Leptos *et al.* PRL 2009) was incomplete : large but rare entrainment events need to be considered to account for the long time dynamics of the beads. These dramatic events could not have been probed by short-time tracking as previously done in [25]. In this sense, only looking at the dynamics of tracers over long times provides the essential feature of their interaction with microorganisms. Of course overall the tracer dynamics is diffusive, but taking the point of view that this is all one needs to know is reductive. There is much more to be understood by uncovering the actual mechanism leading to the diffusivity enhancement. Beyond biomixing and other microhydrodynamic problems related to tracer/microswimmer suspensions (e.g. how to model/simulate them properly), the precise interaction dynamics is likely to be of importance e.g. during feeding of predatory swimming microorganisms, fundamental in the ecology of marine ecosystems. Therefore we strongly believe that there is clear and important merit in showing for the first time that direct entrainment dominates the interaction of these swimming microorganisms with small particles.

Coming to the actual mechanism of entrainment itself, this mechanism is not counterintuitive considering that it has already been experimentally reported on similar active systems that turbulence can lead to a significant enhancement in diffusion (see, e.g., Zhang et al., Europhys Lett 87, 48011 (2009)). Moreover, the data and the analysis performed by the authors are purely phenomenological and do not offer any deep insight on the actual physical mechanism itself, which would be more interesting if assessed experimentally. Is entrainment due to hydrodynamic coupling as suggested? Or are the algae simply pushing the colloids head-on?

The significant enhancement of tracer diffusion seen in Zhang et al., Europhys Lett 87, 48011 (2009) is due to bacterial swarming. This is a well known type of collective motility characteristic of a dense bacterial suspension, where the cells display large scale correlated motion. This system is very far from the one we study in our experiments. We always operate at very low concentrations, where no collective effects can take place : the results we describe are consequences of interactions between tracers and single cells. This is supported e.g. by the linear scaling of we observe for the effective diffusivity on cell concentration. Notice that even for larger CR concentrations, we would not expect the emergence of collective effects similar to those reported in Zhang EPL 2009, since those are predicted to arise only for pusher-type microorganisms (like bacteria), while CR is a puller-type microswimmer. Furthermore, although the trajectories shown in Zhang et al are not analysed in great detail, they look very different from what we report. Indeed, they seem consistent with a persistent random walk, commonly seen in bacteria-enhanced diffusion (see e.g. Wu and Libchaber PRL 2009). The tracer dynamics that we observe presents characteristics that, to the best of our knowledge, have not been reported previously. In particular, it is not a persistent random walk. As we show in the paper, it can instead be described very well by a jump-diffusion process, for which we provide a microscopic explanation (i.e. the existence of rare entrainments).

Regarding the Referee's comment on the entrainment mechanism, our opinion is that there is a fundamental difference between "phenomenological" and "descriptive". Describing a phenomenon quantitatively is the fundamental first step towards its understanding, and this is precisely what we do in the paper, linking the observed enhanced diffusivity (*much* larger than what was previously reported for the same system) with the basic mechanism at the microscopic level (i.e. the existence of rare but large entrainment events). We share with the Referee the opinion that it will be worthwhile in the future

to explore from a near-field microhydrodynamic perspective the mechanisms responsible for the novel phenomenon we report. However, this is clearly beyond the scope of the present paper. We believe that the present paper already presents a strong and important message of broad interest which deserves publication in a major journal like Nat Comm. This is an opinion shared explicitly by both Referees 1 and 2.

- Considering the different types of experiments (i.e. sedimentation, spreading, quasi-2D tracking) that were tried and whose data compared, it would really be worth including detailed schematics of the different setups to help visualize them.

We have taken the Referee’s suggestion into account and have replaced, in the Supplementary Informations, the 2D drawings of the microfluidic chips by detailed schematics of the setups. We hope that these schematics will help understanding better the different setups of the experiments.

- I am a bit puzzled by the authors’ choice to express the enhancement in diffusion as $D_{eff} - D_0$ rather than D_{eff}/D_0 (in Fig. 1, for example).

The quantity most directly related to the activity of the microalgae is what could be called the “excess” diffusivity, i.e. $D_{eff} - D_0$. This is what we are showing in Fig. 1, and it provides an immediate comparison of the slopes α describing the dependence of the effective diffusivities on algal concentration. Furthermore, plotting $D_{eff} - D_0$ shows immediately that the slopes α are proportional to the average swimming speed of the algae. We now know that this fact is due to a simple scaling of the encounter rate between the particles and the swimmer, but it would have been obscured in a plot of D_{eff}/D_0 , due to the fact that the equilibrium diffusivity D_0 is different for the media with/without Percoll.

- The authors compare diffusion coefficients between bulk experiments, quasi-2D experiments, and, as far as I can tell, quasi-1D experiments in the spreading case (schematics would help to understand the actual configuration better), but, at least looking at the equations in the supplementary information, they do not seem to take dimensionality into account when comparing the different values of diffusivity. If dimensionality is not taken into account, the different experiments cannot be directly compared unless opportunely normalized.

At equilibrium the diffusivity D of identical particles performing a random walk is the same whatever the dimension of the supporting medium. The only difference lies in the statistics of displacement which are affected by dimensionality. This is apparent for instance in the second moment of displacements : $\langle \delta r^2 \rangle_d = 2dDt$, where d is the dimension.

In this regard, the quantities presented in Fig. 1c) are true effective diffusivities, obtained by considering the dimensionality of the system. For instance the diffusivities in the quasi-2D experiments have been obtained from the (2D) MSD of the beads $\langle \Delta r^2 \rangle = 4D_{eff}t$. We apologise for the misunderstanding possibly caused by us not being explicit enough on this point. We have now amended the Supplementary Informations to make it clearer.

- Related to the previous point, Ref. [26] reported that a reduced dimensionality leads to a major enhancement of diffusion, while the authors seem to see no such difference. Why is that ?

Within this context, reduced dimensionality means confining the suspension of swimmers and tracers to a thin layer of fluid, realising a quasi-2D system rather than a truly 2D one. This difference might seem just a matter of semantics, but it is in fact very important for the point at hand. The two most straightforward ways to study microswimmers in quasi-2D confinement are either suspending the

system in a soap film, or sandwiching it between two hard surfaces, making what is commonly referred to as a Hele-Shaw cell. Ref. [26] studied the former, while we present results on the latter. In both cases, the microscopic flows generated by microorganisms are profoundly influenced by the presence of boundaries on the top and bottom surfaces of the film. However, the boundary conditions for the fluid are completely different in the two cases : zero normal stress at the fluid-air interface in the soap film, and zero velocity (i.e. no slip) at the fluid-wall boundary in the Hele-Shaw cell. In the soap film case of Ref [26] the stress-free boundaries imply very strong flows from the microorganisms, decaying with distance r from the swimmer as $1/r$. These are much stronger than the flow fields present in bulk, which are known to decay as $1/r^2$ (see Drescher *et al.* PRL 2009), and certainly stronger than those within a Hele-Shaw cell, which will decay at least as fast as in bulk (see Brotto, Caussin, Lauga, and Bartolo 2013) and possibly even faster (alas, these have not been measured yet).

As already stated in the reply to the first point raised by the Referee, the enhancement of diffusion reported in [26] is due to the strong and long-ranged flow fields peculiar of a soap-film geometry.

In our case there is no long-range coupling in any of the three types of experiments performed, regardless of the suspension being in bulk (thickness of the sedimentation channel $\simeq 200 \mu\text{m}$) or in the Hele-Shaw cell ($26 \mu\text{m}$ -thick). Notice that the confinement in our quasi-2D experiments is not very strong (CR cell body diameter $\sim 10 \mu\text{m}$). As a consequence, while we can expect a screening of the far flow field, the near field should be identical to the bulk case, and therefore lead to the same type of entrainment. Given that entrainment dominates the microorganism-tracer interactions, we would then expect to measure the same diffusivity in all our experiments. This is indeed confirmed experimentally.

Of course, if confinement enhances drastically the far flow field of the microorganism then the interaction can change completely from that found in bulk. This, however, is peculiar to the special case of a soap film suspension of microorganisms.

- I wonder why the authors did not also use Percoll in the sedimentation experiment in order to have same viscosity and data directly comparable to the other experiments ?

The Percoll solution used in the spreading and quasi-2D experiments was made to match the density of the beads for the reasons given in the text and Supplementary Informations. We could not use this solution for the sedimentation experiment simply because there would have been no sedimentation to look at for density matched colloids, regardless of the algal concentration (even at $N_c = 0$). In other words, using Percoll for the sedimentation experiments would have prevented us from measuring any (effective) gravitational length, and hence we could not have extracted the effective diffusivities of the beads. Of course, the drawback of this choice is that the medium viscosity is slightly different in the sedimentation experiment than in the two other experiments (we measured $\eta_{\text{percoll}} = (1.5 \pm 0.1)\eta_{\text{water}}$). This difference in viscosity has clearly an effect in the thermal diffusivity of the beads (D_0) and in the swimming speed of the algae. However, it has no effect on the entrainment dynamics besides the trivial decrease of the frequency of entrainment events due simply to the slower swimming of the microorganisms. This conclusion is supported strongly by the fact that the effective diffusivities extracted from the three experiments are identical once time is rescaled by the appropriate ratio of algal swimming speeds.

- In the sedimentation experiment, the authors compare the diffusion calculated from the effective gravitational length and that calculated from short-duration tracking of microparticles and find different values for it. First, can they exclude the effect of drift due to gravity in the first measurement ?

The short-time tracking done in the sedimentation experiment was performed on planes orthogonal to the gravitational field, and hence is not affected by gravitational drifts.

- Then, if no drift is at play, this difference could be explained also by the fact that the two

measurements probe different timescales. For active particles (or colloids in active baths) the diffusive dynamics typically show three different timescales : at very short time scales, these dynamics are diffusive following standard thermal diffusion, at intermediate time scales, they become superdiffusive (growing quadratically in time), and finally they go back to linear diffusion with an enhanced diffusion coefficient. Considering that the tracking was done at short-duration intervals, couldn't it be that the two sets of measurements are either probing intermediate or long-time scales? If that's the case, then the difference in diffusivity is not surprising.

The short-time tracking is done on timescales of a few seconds, the same as those used in Leptos *et al.* PRL 2009 (average track duration 2.6s, now included in the main paper). From the analysis of these tracks we recover the same results obtained in that paper, including a linear mean square displacement and an enhanced diffusivity which varies with cell concentration with the same proportionality coefficient. We do not see any unequivocal evidence of a ballistic regime, and this was the case also in Leptos *et al.* PRL 2009. As was clear in that paper, this enhancement is due to the effect of loops. Since Leptos *et al.* PRL 2009, the idea that loops are the dominant feature describing the effect of microswimmers on small tracers has been discussed theoretically but overall widely accepted, given the experimental support. What we see is that the enhancement due to loops is in fact negligible compared to the effect of entrainments, whose importance, however, becomes evident only at sufficiently long times. We believe this to be an interesting and surprising result, and this opinion is shared also by Referees 1 and 2. Our analysis at the end of the ‘‘Analytical model’’ section, rationalises this fact, and provides an explanation for the reason why using tracks that are just a few seconds long will never provide the correct long-time diffusivity.

- In Fig. 3, the authors fit the distribution of jumps with a Poisson distribution. This fitting seems quite good at long jumps; however, it seems to be quite off when looking at short jumps that are also the most probable ones. I wonder if the Poisson distribution is actually the most appropriate distribution to describe the observed microscopic dynamics?

The distribution of jumps in Fig.3 is actually fitted -above the threshold L_T - with an exponential distribution. Below L_T we are not entirely confident in our ability to distinguish between jumps and loops, and therefore we decided to err on the side of caution and implement in the simulations a hard thresholding at L_T . This is explained in the text and in greater detail in the Supplementary Informations. Our numerical study shows that the short-length jumps, which are not considered in the simulations but are obviously present in the real dynamics, have in fact a negligible effect. Of course an exponential distribution of jump lengths can be thought of as originating from a Poisson process describing the probability of an entrained particle to fall *out* of entrainment. We did in fact use this in the ‘‘Analytical model’’ section of the paper, and it eased its explicit solution. However, whether the jump-length distribution is in fact derived from a Poisson process or not is something that relates to what is the microhydrodynamic origin of the phenomenon, which merits a dedicated study beyond the scope of the present paper.

A different process describes the transition of a colloid from the diffusing to the entrained state. We see a distribution of waiting times between successive entrainments which is well characterised by an exponential (e.g. Fig. 3b inset), with characteristic times that are inversely proportional to cell concentration (Fig.3b main panel). These observations support strongly the conclusion that this transition dynamics is a simple Poisson process with a rate that is proportional to CR concentration, at least for the concentrations we looked at. This is the Poisson process that is found in Eq.(1), and it is the Poisson process we refer to when saying that the colloids undergo a jump-diffusion process with Poisson-distributed jumps.

Reviewer #3 (Remarks to the Author):

First of all, I would like to thank the authors for clarifying the difference between their work and reference [26] that was not clear to me in the first iteration. Considering this key difference, I agree with the authors that this finding, if confirmed, deserves in principle to appear in an outlet, such as Nature Communications.

We thank the Reviewer for the supportive remarks. At the same time, we find the comment on the need to further confirm our results to be completely unwarranted, given the **quantitative** internal consistency of all of our experiments.

Our results from the sedimentation experiments show unequivocally that the previous measurements of the effective diffusivity in the same geometry (Leptos 2009) were largely underestimating the effect of microalgae on suspended particles. The fact that our short timescale *in situ* microscopic measurements of the diffusivity within the sedimentation experiments return a value compatible with Leptos 2009 shows that a key element of the microparticle dynamics is missing from the short-timescale particle tracks. We then re-measured the macroscopic diffusivity by looking at the dynamics of spreading, and again found results that are quantitatively perfectly compatible with the macroscopic sedimentation experiments. We stress that spreading and sedimentation experiments provide diffusivity measurements which are **conceptually and procedurally** completely different from each other. Their remarkable agreement is a solid confirmation of the macroscopic results. To investigate the discrepancy between these measurements and those from short-timescale particle tracking, we have performed experiments in a Hele-Shaw geometry, which allows us to track the particles for a long time. The resulting dynamics reveals the existence of rare particle entrainment, and shows that this is the missing part of the dynamics. In fact, the diffusivity we measure from the 2D projection of the particle trajectories is in clear quantitative agreement with the one we measure in the macroscopic experiments. The simulations and modelling confirm that entrainments are the most important components of the dynamics and rationalise why long-timescale trajectories are needed to correctly sample the dynamics. Finally, one of our Supplementary Movies shows direct evidence of entrainments also in the sedimentation experiments.

Overall, our results provide an account of microparticles' dynamics in these active systems which is derived from fundamentally different types of measurement (macroscopic steady state; macroscopic spreading; microscopic) all of which agree **quantitatively**. Therefore the results have already been thoroughly tested and repeatedly confirmed in independent measurements.

Regarding the comments on the effect of dimensionality on the effective diffusion coefficient (as reported in Ref. [26]), this effect seems to be an intrinsic characteristic of particles that show an enhanced diffusivity. While for Brownian particles the diffusion coefficient does not depend on the dimensionality of the system as the authors correctly point out in their reply, this seems not to be the case for systems with enhanced diffusion, as for example, reported in this theoretical work: M. E. Cates and J. Tailleur, EPL, 2013, 101, 20010. I remain therefore puzzled by the fact that the authors do not observe this effect at all. Can they please comment on this? Generally speaking, I think that adding some explanations about the dependence of the diffusivity on the dimensionality in the manuscript would be helpful for the general reader.

We agree that the effect of dimensionality on active or mixed active/passive systems is indeed an interesting and worthwhile topic to study. The first case -purely active suspensions- is investigated theoretically and with simulations e.g. in the paper suggested by the Reviewer. Instead, in that paper we did not see any reference on active/passive systems like ours. Of course we agree that the effect of dimensionality on enhanced diffusion of passive particles is certainly a topic ripe for further progress, at least from a simulation/theory perspective. However, we do not believe that this is something that should be addressed by the present manuscript. It is certainly not its focus, and

indeed **we do not claim at all in the paper** that the diffusivity would be the same in the purely 2D case -whatever this means experimentally-

In the Hele-Shaw experiments we look at the 2D projection of the particle coordinates, and we use the confinement as an expedient to track microparticles for a long time. We have now made this point more explicit in the text: see list of changes below. The dynamics projected on the field of view has a diffusivity which is **the same** as that measured macroscopically, and shows furthermore that this enhanced diffusivity results from entrainment events, not previously reported. The diffusivity agreements and the fact that we observe entrainments also in the sedimentation experiments (see Supplementary Movie) supports our claim that the dynamics we observe in the Hele-Shaw experiments is representative of the general dynamics in 3D, which is the focus of the paper. To avoid confusion we have amended the text to avoid referring to the experiments in the Hele-Shaw cell as “quasi-2D”. They are now called either “confined” or “Hele-Shaw” experiments.

Text changes:

1) On page 3 we substituted

“[...]The crucial microscopic insight is provided by long-time tracking of the particles, here individually followed for ~ 200 s within a $26\mu\text{m}$ -thick Hele-Shaw cell, at a range of N_c values. [...]”

with

“[...] The crucial microscopic insight is provided by long-time tracking of the particles at a range of N_c values. This is achieved here by confining the system within a $26\mu\text{m}$ -thick Hele-Shaw cell, which allows us to follow individual colloids for ~ 200 s. [...]”

2) On page 5 we substituted

“[...]Consequently, we model the stochastic trajectory of a colloid within a quasi-2D suspension of microalgae, [...]”

with

“[...]Consequently, we model the 2D projection of the stochastic trajectory of a colloid in the Hele-Shaw experiments [...]”